# Clinicopathological Features and Significance of Epidermal Growth Factor Receptor Mutation in Surgically Resected Early-Stage Lung Adenocarcinoma

**DOI:** 10.3390/diagnostics13030390

**Published:** 2023-01-20

**Authors:** Chao-Wen Lu, Mong-Wei Lin, Xu-Heng Chiang, Hsao-Hsun Hsu, Min-Shu Hsieh, Jin-Shing Chen

**Affiliations:** 1Department of Surgery, National Taiwan University Hospital and National Taiwan University College of Medicine, Taipei 100, Taiwan; 2Graduate Institute of Pathology, National Taiwan University College of Medicine, Taipei 100, Taiwan; 3Department of Medical Education, National Taiwan University Hospital and National Taiwan University College of Medicine, Taipei 100, Taiwan; 4Department of Pathology, National Taiwan University Hospital, Taipei 100, Taiwan; 5Department of Pathology, National Taiwan University Cancer Center, Taipei 100, Taiwan; 6Department of Surgery, National Taiwan University Cancer Center, Taipei 100, Taiwan

**Keywords:** lung adenocarcinoma, EGFR, exon 19 deletion, L858R

## Abstract

The clinicopathological presentation of early-stage lung adenocarcinoma patients with epidermal growth factor receptor (EGFR) mutations has been seldom studied. Our study enrolled patients with stage I and II lung adenocarcinoma between January 2014 and December 2017 at the National Taiwan University Hospital. Clinicopathological features and prognosis were retrospectively reviewed and analyzed depending on EGFR mutation status. EGFR mutations were detected in 622 (60%) out of 1034 patients. Compared to the group without EGFR mutations, the group with EGFR mutations had more patients above 65 years of age (*p* < 0.001), more non-lepidic histological subtypes (*p* < 0.001), higher CEA levels (*p* = 0.044), higher grade of pleural (*p* = 0.02) and lymphovascular (*p* = 0.001) invasion, higher histological grade (*p* < 0.001), and a more advanced pathological stage (*p* = 0.022). In multivariate analysis, there was no significant difference in PFS or OS between the EGFR mutant and wild-type groups. In subtype analysis, the tumors with an L858R mutation had a more lepidic predominant histological type (*p* = 0.019) and less lymphovascular invasion (*p* = 0.011). No significant differences in PFS or OS were detected between the exon 19 deletion and L858R mutation groups. In early-stage lung adenocarcinoma, EGFR mutation may be considered as a treatment response predictor for tyrosine kinase inhibitors, instead of a predictor of clinical prognosis.

## 1. Introduction

Epidermal growth factor receptor (EGFR) mutation is a predictor of responsiveness to lung-adenocarcinoma-targeted drug tyrosine-kinase inhibitor (TKI) therapy [1,2]. According to current treatment guidelines for lung adenocarcinoma, detection of EGFR mutations is the first recommendation for patients with advanced or inoperable lung adenocarcinoma [3,4]. For patients with early-stage lung adenocarcinoma, curative surgery is the standard treatment [5,6,7,8]; some studies have pointed out that patients with early-stage lung adenocarcinoma with EGFR mutations have a lower recurrence rate after surgery [9,10]. There are some studies on the pathological analysis of EGFR mutations in patients with early-stage lung adenocarcinoma [11,12].

The impact of the EGFR mutations on operable non-small-cell lung cancer (NSCLC) has been evaluated in different reports. In some studies, EGFR mutations presented as an improved prognostic factor for recurrence rate or overall survival even in operable NSCLC [13,14,15], while others showed that EGFR mutations are not a prognostic factor in early-stage NSCLC [11,16]. There is increasing interest in the relationship between EGFR mutations and resectable lung adenocarcinoma [16]. The ADAURA trial showed longer disease-free survival (DFS) in patients with EGFR-mutant stage IB-IIIA NSCLC who received osimertinib after surgery [3,17,18]. In Spain, a Delphi consensus panel suggested that the EGFR mutation test should be performed after surgery in patients with early-stage NSCLC [19]. Thus, the importance of EGFR mutations in early-stage NSCLC prognosis has gained recognition in recent years.

This study aimed to analyze the postoperative prognosis and clinicopathological characteristics associated with EGFR mutations in operable lung adenocarcinoma. The common drug-treatable EGFR mutations (Exon 21 L858R point mutation and Exon 19 deletion) were also examined.

## 2. Materials and Methods

### 2.1. Study Population

Medical records were reviewed retrospectively to select patients with pathological stage I or II lung adenocarcinoma, who underwent pulmonary resection at the National Taiwan University Hospital between January 2014 and December 2017. A total of 2064 patients received pulmonary resection between January 2014 and December 2017; among them, 1915 patients were diagnosed with lung adenocarcinoma. A total of 1658 patients had stage I or II lung adenocarcinoma (based on the American Joint Committee on Cancer (AJCC) 8th edition TNM staging system for lung cancer). The EGFR gene mutation test was conducted on 1034 specimens from stage I or stage II patients. EGFR mutations were detected in 622 (60%) patients. Figure 1 shows the flowchart of patient enrollment. This study was approved by the Institutional Research Ethics Committee (approval no. 202006178RINB). The requirement for informed consent was waived by the committee due to the retrospective nature of the study.

Electronic medical records of the enrolled patients were collected for sex, age at operation, smoking history, family history of lung adenocarcinoma, preoperative serum carcinoembryonic antigen (CEA) level, pathological tumor size, presence of a lepidic growth pattern, lymphovascular and pleural invasion status, T status, N status and stage, and adjuvant treatment status. Histological classification and pathological features were classified according to the 2021 World Health Organization classification of thoracic tumors [20].

### 2.2. Management and Follow-up

After operation, pathological stage IA patients received regular clinic follow-up for 5 years, while stage IB patients received further adjuvant therapy after discussion with the tumor board. Stage II patients were referred to the medical oncologist for further adjuvant chemotherapy.

Follow-up assessments included physical examinations, blood tests including CEA levels, and chest computed tomography (CT) scans every 6 months for 5 years. If a patient showed symptoms or signs of recurrence, further examinations such as positron emission tomography, brain CT, or brain MRI were performed. The diagnosis of recurrence was confirmed through imaging evidence and/or pathological evidence from tissue biopsy. The disease-free survival (DFS) was defined as the interval between the date of confirmed recurrence and the operation date. The overall survival (OS) was defined as the length of time from a patient’s operation until death.

### 2.3. Analysis of EGFR Mutation

A formalin-fixed, paraffin-embedded (FFPE) sample from the resected tumor was used for analysis of the EGFR mutations. DNA was extracted using an FFPE NA Extraction Kit (SPRI-TETM Nuclei Acid Extractor). The quality and quantity of DNA were evaluated using a NanoDrop (ND-1000). EGFR mutations were detected using Mass ARRAY genotyping (SEQUENOM) as previously described [21].

### 2.4. Statistical Analysis

Categorical variables are presented as numbers (percentages), and descriptive statistics are shown as means ± standard deviations. The chi-squared test was used for categorical variables, and the Student’s *t*-test was performed for continuous variables. Disease-free survival (DFS) and overall survival (OS) were evaluated using the Kaplan-Meier method and log-rank test. The Cox regression model was used for multivariate analysis of the significant confounding factors noted in the univariable model. Statistical significance was set at *p* < 0.05. Statistical analyses were performed using the SPSS software version 25 (IBM Corp., Armonk, NY, USA).

## 3. Results

### 3.1. Patient Demographics and Clinicopathological Characteristics

The clinicopathological characteristics of EGFR mutation status are shown in Table 1. Compared to the group without EGFR mutations, the group with EGFR mutations had a greater number of patients above 65 years of age (*p* < 0.001) and had higher CEA levels (*p* = 0.044). More non-lepidic histological subtypes were observed in the EGFR mutation group (*p* < 0.001). Pleural (*p* = 0.009) and lymphovascular (*p* = 0.001) invasion, higher histological grade (*p* < 0.001) and T-stage (*p* < 0.001), and a more advanced pathological stage (*p* = 0.022) were observed in the EGFR mutation group. There was no difference in sex, smoking, or nodal status between the EGFR-positive and -negative groups.

### 3.2. Clinical and Pathological Characteristics of Tumors with EGFR Exon 19 Deletion and L858R Mutation

The differences in clinicopathological characteristics of EGFR Exon 19 deletion and exon 21 L858R point mutation are presented in Table 2. Among 621 patients with EGFR mutations, 241 (38.9%) had exon 19 deletions and 306 (49.3%) had exon 21 L858R point mutations. Compared to the exon 19 deletion group, patients with the L858R mutation had a higher smoking history (*p* = 0.006). Tumors with the L858R mutation were found to have a more lepidic predominant histology subtype (*p* = 0.019) and less lymphovascular invasion (*p* = 0.011).

### 3.3. Survival Analysis

Table 3 shows the results of univariate analysis and multivariate analysis of DFS. In the univariate analysis, the associated factors were as follows: age > 65 years (*p* = 0.002), smoking history (*p* = 0.016), CEA level (*p* < 0.001), tumor size (*p* < 0.001), lepidic growth pattern in the tumor (*p* = 0.004), presence of pleural (*p* < 0.001) and lymphovascular (*p* < 0.001) invasion, N1 lymph node metastasis (*p* < 0.001), and EGFR mutations (*p* = 0.023). With multivariant analysis, age > 65 years (HR = 1.508, 95% CI = 1.032–2.203, *p* = 0.034), CEA level (HR = 1.895, 95% CI= 1.204–2.982, *p*= 0.006), tumor size (HR = 2.494, 95% CI= 1.574–3.953, *p* < 0.001), presence of lymphovascular invasion (HR = 1.742, 95% CI = 1.118–2.714, *p* = 0.014), and N1 lymph node metastasis (HR = 2.055, 95% CI = 1.282–3.293, *p* = 0.003) were independent factors.

Table 4 shows the results of univariate and multivariate analysis of OS. In univariate analysis, tumor size (*p* = 0.01), presence of pleural (*p* = 0.002) and lymphovascular (*p* < 0.001) invasion, and N1 lymph node metastasis (*p* < 0.001) were associated factors. Smoking habit, lepidic growth pattern, presence of pleural invasion, and EGFR mutations were not independent risk factors for disease recurrence after surgery. In the multivariate analysis, the presence of lymphovascular invasion (HR = 3.744, 95% CI = 1.175–11.937, *p* = 0.026) and N1 lymph node metastasis (HR = 3.719, 95% CI = 1.145–12.076, *p* = 0.029) were associated with poor OS.

The Kaplan–Meier curves for DFS and OS in the EGFR mutation subtypes of exon 19 deletion, L858R and EGFR wild-type are presented in Figure 2 and Figure 3, respectively. No significant differences in DFS and OS were detected (*p* = 0.078 and *p* = 0.932, respectively).

## 4. Discussion

In our study population, 60% of patients had EGFR mutations. This is similar to the results of previous studies [11,22]. Our study showed that the EGFR mutation group was associated with aggressive clinicopathological features such as older age, non-lepidic histological subtype, higher rate of pleural and lymphovascular invasion, higher serum CEA level, tumor histological grade, and T stage, and more advanced pathological stage. Other studies did not show a significant difference in age or serum CEA levels [9,14,23]. A previous study by Yotsukura et al. showed a lower level of serum CEA in the EGFR mutant group [11]. Several studies have reported a more lepidic pattern of lung adenocarcinoma in patients with EGFR mutations [11,23,24] which is not consistent with our study. However, a study by Nie et al. demonstrated higher pleural invasion in the EGFR mutation group [25]. Deng et al. found that the EGFR mutation group had a higher histological grade than wild-type patients [26]. We also noted that the minimally invasive adenocarcinoma group had lower EGFR mutation rates. This result is similar to that reported by Haiquan et al. [27]. All these clinicopathological features might represent a more locally invasive pattern of the EGFR mutation group in early-stage lung adenocarcinoma.

In our study, exon 19 deletion and exon 21 L858R point mutation accounted for 88% of the EGFR mutation group, which is similar to the results of previous studies [11,28]. A study by Yotsukura et al. found no difference between the exon 19 deletion and L858R groups in smoking habits or lymphovascular invasion status. However, the L858R mutation group tended to have more tumors with lepidic growth patterns [11]. In our study, which enrolled more patients than that of Yotsukura et al., we found more significant difference that the L858R mutation group had more tumors with lepidic growth patterns. Many studies have shown a better treatment effect of exon 19 deletion than L858R mutation group in advanced stage lung cancer treated with TKI therapy [28,29,30,31,32]. In our study, there was no significant difference between the exon 19 deletion and L858R mutation groups, which might have been due to the enrollment of only early-stage cases in our study. This result is similar to that of a study by Yotsukura et al. [11].

Few studies have demonstrated the treatment response of patients with uncommon EGFR mutations. Patients with G719X, S768I, or L861Q mutations responded to TKI, while those with exon 20 insertions were mostly TKI-insensitive [33,34].

Several studies have revealed that DFS and OS are affected by lymphovascular invasion and nodal metastasis status [13,35,36]. This finding is consistent with the results of our study. The difference in the prognosis of early-stage operable lung adenocarcinoma between EGFR mutation and wild-type patient groups was inconclusive in previous studies. Some studies have shown that EGFR mutations are a better prognostic predictor in cases of operable non-small cell lung cancer [9,14], while others have revealed no significant impact of EGFR mutations [16,35,37]. Our study showed that EGFR mutations are not a prognostic factor for patients with early-stage lung adenocarcinoma. In addition, the prognosis between the exon 19 deletion and L858R mutation groups was not consistent with previous studies. Some studies showed no significant differences in prognosis between patients with exon 19 deletion and L858R mutation [11,14], which is consistent with our study. However, Li et al. reported better survival outcomes in the exon-19-deletion group [37].

Invasive pathological features such as lymphovascular invasion and nodal metastasis have been shown to be associated with poor DFS and OS [13,35]. In this study, we did observe that non-lepidic pattern, pleural invasion, the presence of lymphovascular invasion, higher histologic grading, higher T1 stage (T1a-c), and higher stage IB were significantly more common in EGFR mutation-positive patients than those lacking mutations. Nevertheless, only tumor size, lymphovascular invasion, and nodal metastasis were found to be significant factors in multivariate analysis. The presence or absence of EGFR mutations was not found to be a significant factor in the prediction of PFS (*p* = 0.772) and OS (*p* = 0.430). This finding is consistent with a recently published French study [16] that found that EGFR mutations were not associated with the recurrence site, disease-free survival, or overall survival in resected stage I–II NSCLC.

Our study had several limitations. Firstly, this was a retrospective study and bias may have been present. Secondly, although a large number of patients were enrolled from a single team, the results may not present the patient characteristics thoroughly. Thirdly, only 62% patients with stage I or II EGFR mutations might not show the complete picture of clinicopathological characteristics of patients with or without EGFR mutations.

## 5. Conclusions

In stage I and stage II operable lung adenocarcinoma, our findings suggest that EGFR mutations may be considered as a treatment response predictor for TKI, and may not be a predictor of clinical prognosis. The results of our study should be further validated by other multi-institutional studies.

## Figures and Tables

**Figure 1 diagnostics-13-00390-f001:**
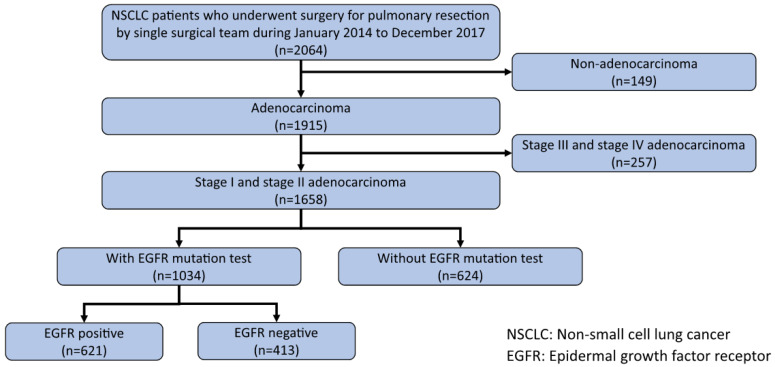
Flow chart of patient enrollment.

**Figure 2 diagnostics-13-00390-f002:**
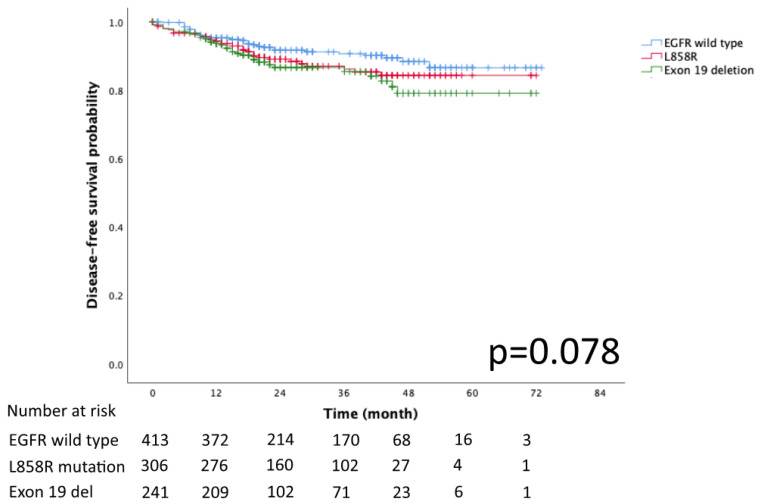
Kaplan–Meier curve of EGFR wild type, Exon 19 del, and L858R disease free survival.

**Figure 3 diagnostics-13-00390-f003:**
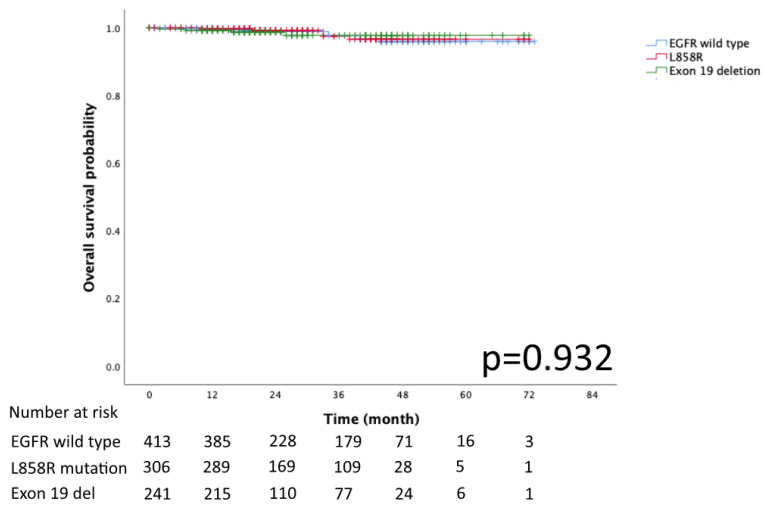
Kaplan–Meier curve of EGFR wild type, Exon 19 deletion, and L858R overall survival.

**Table 1 diagnostics-13-00390-t001:** Patients clinicopathological characteristics.

Variables	EGFR(+) n = 621	EGFR(−) n = 413	*p*-Value
Sex, n (%)			0.893
Female	411 (66.2)	275 (66.4)	
Male	210 (33.8)	138 (33.6)	
Age			<0.001
≤65	365 (58.7)	299 (72.5)	
>65	256 (41.3)	114 (27.5)	
Smoking, n (%)			0.137
No	538 (86.7)	344 (83.1)	
Yes	83 (13.3)	69 (16.9)	
Family lung cancer history			0.307
No	499 (80.4)	321 (77.8)	
Yes	122 (19.6)	92 (22.2)	
CEA level			
≤5 ng/mL	549 (88.3)	381 (92.3)	0.044
>5 ng/mL	72 (11.7)	32 (7.7)	
Histologic predominant subtype, n (%)			<0.001
Lepidic predominant	52 (8.4)	102 (24.6)	
Non-lepidic predominant	569 (91.6)	311 (75.4)	
Pleural invasion, PI, n (%)			0.02
PL0	510 (58.4)	364 (88.2)	
PL1	66 (10.6)	41 (9.9)	
PL2	39 (6.3)	7 (1.7)	
PL3	6 (0.9)	1 (0.2)	
Lymphovascular invasion, n (%)			0.001
Absent	504 (81.2)	367 (88.9)	
Present	117 (18.8)	46 (11.1)	
Histologic grade, n (%),			<0.001
1	135 (20.9)	164 (39.4)	
2	386 (59.8)	198 (46.1)	
3	98 (14.1)	48 (11.1)	
T stage, n (%)			<0.001
T1mi	23 (4.0)	68 (16.5)	
1a	109 (17.6)	134 (32.4)	
1b	189 (30.4)	92 (22.3)	
1c	123 (19.8)	34 (8.2)	
Stage 2 and 3	177 (28.5)	85 (20.6)	
LN metastasis			0.165
N0	557 (86.2)	381 (88.1)	
N1	64 (13.8)	32 (11.9)	
Pathological stage, n (%)			0.022
IA	440 (70.9)	327 (79.2)	
IB	141 (22.7)	63 (15.2)	
IIA	16 (2.6)	9 (2.2)	
IIB	24 (3.9)	14 (3.4)	

CEA: carcinoembryonic antigen. T1mi: minimally invasive adenocarcinoma. LN: lymph node

**Table 2 diagnostics-13-00390-t002:** Clinicopathological differences between Exon 19 deletion and L858R mutation.

Variables	Del-19 (n = 241)	L858R (n = 306)	*p*-Value
Sex, n (%)			0.076
Female	152 (63.2)	215 (70.3)	
Male	89 (36.8)	91 (29.7)	
Age			0.185
≤65	149 (61.6)	172 (56.2)	
>65	92 (38.4)	134 (43.8)	
Smoking, n (%)			0.006
Absent	198 (82.2)	276 (90.2)	
Present	43 (17.3)	30 (9.8)	
Family lung cancer history			0.617
Absent	194 (80.6)	241 (78.8)	
Present	47 (19.4)	65 (21.2)	
Preoperative CEA > 5 ng/mL, n (%)			0.530
Absent	210 (96.0)	272 (96.9)	
Present	31 (4.0)	34 (3.1)	
Histologic predominant subtype, n (%)			0.019
Lepidic predominant	12 (5.0)	32 (10.5)	
Non-lepidic predominant	229 (95.0)	274 (89.5)	
Pleural invasion, PI, n (%)			0.555
PL0	199 (82.6)	249 (81.4)	
PL1	24 (10.0)	33 (10.8)	
PL2	17 (7.0)	19 (6.2)	
PL3	1 (0.4)	5 (1.6)	
Lymphovascular invasion, n (%)			0.011
Absent	185 (76.9)	261 (85.3)	
Present	56 (23.1)	45 (14.7)	
Histologic grade, n (%),			0.424
1	58 (24.1)	66 (21.6)	
2	141 (58.5)	196 (64.1)	
3	41 (17.0)	43 (14.1)	
LN metastasis			0.097
N0	210 (87.1)	280 (91.5)	
N1	31 (12.9)	26 (8.5)	
Pathological stage, n (%)			0.457
IA	171 (71.1)	214 (70.0)	
IB	56 (23.1)	72 (23.5)	
IIA	8 (3.3)	6 (1.9)	
IIB	6 (2.5)	14 (4.6)	

**Table 3 diagnostics-13-00390-t003:** Univariate and multivariate analysis of disease-free survival probabilities after surgery.

Variables	Univariate Analysis			Multivariate Analysis		
	HR	95% CI	*p*-Value	HR	95% CI	*p*-Value
Age						
≤65		ref.			ref.	
>65	1.788	1.240–2.577	0.002	1.508	1.032–2.203	0.034
Sex						
Female		ref.			ref.	
Male	0.833	0.571–1.216	0.344	0.942	0.590–1.505	0.804
Smoking						
Absent		ref.			ref.	
Present	1.722	1.106–2.683	0.016	1.251	0.710–2.204	0.438
CEA level						
≤5 ng/mL		ref.			ref.	
>5 ng/mL	4.311	2.832–6.563	<0.001	1.895	1.204–2.982	0.006
Tumor size						
≤ 20 mm		ref.			ref.	
>20 mm	4.828	3.222–7.234	<0.001	2.494	1.574–3.953	<0.001
Histologic predominant subtype						
Non-Lepidic		ref.			ref.	
Lepidic	0.330	0.154–0.709	0.004	0.697	0.314–1.549	0.376
Pleural invasion, PI, n (%)						
Absent		ref.			ref.	
Present	3.505	2.391–5.136	<0.001	1.509	0.992–2.295	0.055
Lymphovascular invasion						
Absent		ref.			ref.	
Present	4.131	2.838–6.014	<0.001	1.742	1.118–2.714	0.014
Lymph node metastasis						
N0		ref.			ref.	
N1	5.303	3.548–7.924	<0.001	2.055	1.282–3.293	0.003
EGFR mutation						
Absent		ref.			ref.	
Present	1.581	1.065–2.346	0.023	1.964	0.698–1.622	0.772

**Table 4 diagnostics-13-00390-t004:** Univariate and multivariate analysis of overall survival probabilities after surgery.

Variables	Univariate Analysis			Multivariate Analysis		
	HR	95% CI	*p*-Value	HR	95% CI	*p*-Value
Age						
≤65		ref.			ref.	
>65	1.253	0.512–3.067	0.621	1.596	0.618–4.119	0.334
Sex						
Female		ref.			ref.	
Male	1.363	0.557–3.336	0.497	1.392	0.500–3.876	0.527
Smoking						
Absent		ref.			ref.	
Present	1.093	0.320–3.730	0.887	1.473	0.364–5.955	0.587
CEA level						
≤5 ng/mL		ref.			ref	
>5 ng/mL	1.515	0.348–6.594	0.580	1.904	0.408–8.884	0.412
Tumor size						
≤20 mm		ref.			ref	
>20 mm	3.359	1.340–8.420	0.010	1.385	0.441–4.350	0.577
Histologic predominant subtype						
Non-lepidic		ref.			ref	
Lepidic	0.888	0.260–3.033	0.850	0.508	0.129–2.000	0.332
Pleural invasion						
Absent		ref.			ref	
Present	4.162	1.699–10.197	0.002	1.895	0.654–5.493	0.239
Lymphovascular invasion						
Absent		ref.			ref	
Present	7.329	3.034–17.703	<0.001	3.744	1.175–11.937	0.026
Lymph node metastasis						
N0		ref.			ref	
N1	8.895	3.684–21.473	<0.001	3.719	1.145–12.076	0.029
EGFR mutation						
Absent		ref.			ref	
Present	0.908	0.376–2.193	0.829	0.680	0.262–1.770	0.430

## Data Availability

Not applicable.

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
