# Peer review of "Clinicopathological Features and Significance of Epidermal Growth Factor Receptor Mutation in Surgically Resected Early-Stage Lung Adenocarcinoma"

_diagnostics, 2023, doi:10.3390/diagnostics13030390_

Round 1
Reviewer 1 Report
Generally, this manuscript has a large amount of data, a clear idea, and detailed analysis among various subgroup. I think that this article can be further improved. (1) "Materials and methods" section had missing clinical follow-up data. The calculation of DFS and OS should be described in more detail in the methods section.(2) whether the patient has received anti-tumor treatment after surgery should also be introduced in detail to make the analysis results more reliable.Author Response
Response to Reviewer 1 Comments
Point 1: "Materials and methods" section had missing clinical follow-up data. The calculation of DFS and OS should be described in more detail in the methods section.
Response 1: Thank you for your kind response and comments, the paragraph was added in “Materials and methods” section 2.2 Management and follow-up:
“Follow-up assessments for patients included physical examination, blood tests including CEA levels, and chest CT every 6 months for 5 years. If the patient showed symptoms or signs of recurrence, further examinations such as positron emission tomography, brain CT, or brain MRI were arranged. The diagnosis of recurrence was confirmed through imaging evidence and/or pathological evidence from tissue biopsy. The disease-free survival (DFS) is defined as the interval between the date confirmed the recurrence and the operation date. The overall survival (OS) is defined as the length of time from a patient’s operation until the death.”
Point 2: whether the patient has received anti-tumor treatment after surgery should also be introduced in detail to make the analysis results more reliable.
Response 2:
After the operation, the pathological stage IA patient will receive 5 years regular clinic follow-up. Stage IB patients will receive further adjuvant therapy after the tumor board discussion. The stage II patients will be referred to the medical oncologist for further adjuvant chemotherapy .

Reviewer 2 Report
This is a review of the manuscript submitted to the journal Diagnostics by Lu and colleagues entitled “Clinicopathological features and significance of epidermal growth factor receptor mutation in surgically resected early-stage lung adenocarcinoma”. Overall, presentation of the data could use minor revisions, as indicated below, however, there are major gaps in the review of the literature and significance and concordance of the study findings in relation to many articles recently published on the topic of EGFR mutations in resectable NSCLC. A recommendation of ‘Reconsider after major revision (control missing in some experiments)’ is therefore made with the hopes of a revised draft being much closer to the caliber of word expected for this journal.
Major points of concern:
There exists a reasonable number of manuscripts in the literature on the topic of EGFR mutations in resectable NSCLC. A brief review of these should be provided to the audience as well as the results of the current study be placed into context and implications for the new findings, if any. Some examples include (provided as PubMed PMIDs): 36180314, 36292235, 36108559, 36168085 (as a few examples).
Minor points of concern
The title and the body of the manuscript should specify “actionable” EGFR mutations and not any. There are many examples of EGFR mutations in the early stage patients that are not actionable, but have been shown to have some prognostic value. Papers on these other mutations should be referenced in the review.
The finding that there are associations of actionable mutations with traits associated with more aggressive disease, but not with clinical outcomes, should be discussed further. What does this mean? Please further relate these finding to other studies and place into a clear and conclusive context.
The T stage in Table 1 is misleading and different from all other descriptions in Tables 1 and 2. The % composition of the individual T stages for a particular subgroup (EGFRm vs EGFRwt) should add up to be 100%, not the combination of the two subgroups for a given stage. With this, the distribution of cases is obscured and bias in the EGFRwt group is present. This should be discussed.
Table 3 has inappropriate comparisons being made in terms of DFS and OS. It is well known increased stage as a result of LN metastases negatively profoundly impacts clinical outcomes in a manner independent from EGFRm status. With this, associations of EGFRm (or EGFRwt) population with clinical outcomes must control for nodal status.
Figures 2 and 3 should also illustrate the EGFRwt populations.
Author Response
Response to Reviewer 2 Comments
Major points of concern: There exists a reasonable number of manuscripts in the literature on the topic of EGFR mutations in resectable NSCLC. A brief review of these should be provided to the audience as well as the results of the current study be placed into context and implications for the new findings, if any. Some examples include (provided as PubMed PMIDs): 36180314, 36292235, 36108559, 36168085 (as a few examples).
Response: Thank you for your comment. We added the brief review of relevant studies in the introduction paragraph.
“The impact of the EGFR mutation in the operable NSCLC has been evaluated in different reports. In some studies, the EGFR mutation presented as a better prognostic factor of recurrence rate or the overall survival even in the operable NSCLC[13-15]. Some studies showed the EGFR mutation is not the prognostic factor in the early-stage NSCLC [11, 16]. There is increasing interest in the relationship between the EGFR mutation and resectable lung adenocarcinoma [16]. The ADAURA trial showed longer disease-free survival (DFS) in patients with stage IB-IIIA EGFR-mutant non-small cell lung cancer (NSCLC) who received the osimertinib after the surgery [3, 17, 18]. In Spain, a Delphi consensus panel suggested the EGFR mutation test should be performed after the surgery of early-stage NSCLC [19]. The importance of EGFR mutation in early-stage NSCLC has gained recognition in recent years.
This study aimed to analyze the postoperative prognosis and clinicopathological characteristics of the EGFR mutations in operable lung adenocarcinoma. The druggable, common EGFR mutations (Exon 21 L858R point mutation and Exon 19 deletion), will also be discussed.”
Reference:
- Wu YL, Tsuboi M, He J, John T, Grohe C, Majem M, Goldman JW, Laktionov K, Kim SW, Kato T et al: Osimertinib in Resected EGFR-Mutated Non-Small-Cell Lung Cancer. N Engl J Med 2020, 383(18):1711-1723.
- Isaka T, Ito H, Yokose T, Saito H, Adachi H, Murakami K, Miura J, Kikunishi N, Rino Y: Prognostic factors for relapse-free survival in stage IB-IIIA primary lung adenocarcinoma by epidermal growth factor receptor mutation status. BMC Cancer 2022, 22(1):966.
- Takamochi K, Oh S, Matsunaga T, Suzuki K: Prognostic impacts of EGFR mutation status and subtype in patients with surgically resected lung adenocarcinoma. J Thorac Cardiovasc Surg 2017, 154(5):1768-1774 e1761.
- Chen YY, Chen YS, Huang TW: Prognostic Impact of EBUS TBNA for Lung Adenocarcinoma Patients with Postoperative Recurrences. Diagnostics (Basel) 2022, 12(10). (PMID: 36180314)
- Mordant P Md P, Brosseau S, Milleron B, Santelmo N, Fraboulet-Moreau S, Besse B, Langlais A, Gossot D, Thomas PA, Pujol JL et al: Outcome of Patients With Resected Early-Stage Non-small Cell Lung Cancer and EGFR Mutations: Results From the IFCT Biomarkers France Study. Clin Lung Cancer 2023, 24(1):1-10. (PMID: 36180314)
- Wu YL, John T, Grohe C, Majem M, Goldman JW, Kim SW, Kato T, Laktionov K, Vu HV, Wang Z et al: Postoperative Chemotherapy Use and Outcomes From ADAURA: Osimertinib as Adjuvant Therapy for Resected EGFR-Mutated NSCLC. J Thorac Oncol 2022, 17(3):423-433.
- Hardenberg MC, Patel B, Matthews C, Califano R, Garcia Campelo R, Grohe C, Hong MH, Liu G, Lu S, de Marinis F et al: The value of disease-free survival (DFS) and osimertinib in adjuvant non-small-cell lung cancer (NSCLC): an international Delphi consensus report. ESMO Open 2022, 7(5):100572. (PMID: 36108559)
- Isla D, Felip E, Garrido P, Insa A, Majem M, Remon J, Trigo JM, de Castro J: A Delphi consensus panel about clinical management of early-stage EGFR-mutated non-small cell lung cancer (NSCLC) in Spain: a Delphi consensus panel study. Clin Transl Oncol 2022. (PMID: 36168085)
The suggested references are marked in red color.
Minor points of concern 1: The title and the body of the manuscript should specify “actionable” EGFR mutations and not any. There are many examples of EGFR mutations in the early stage patients that are not actionable, but have been shown to have some prognostic value. Papers on these other mutations should be referenced in the review.
Response 1:
Thank you for your comment. Our only enrolled stage I and II lung cancer patients from 2014 to 2017 and none of them received EGFR TKIs after the surgery. We listed all types of EGFR mutations in Table 1, Table 3, and Table 4 to show the detailed EGFR mutation profiles of this large cohort of early lung cancer in Taiwan. The classic EGFR mutations (L858R and Exon 19 deletion) account for 88% of EGFR mutation-positive cases. According to your kind suggestion, DFS and OS of stage I/II lung cancers with wild-type EGFR, classic EGFR mutations, and uncommon EGFR mutations were compared and there were no significant differences among these three groups.
The patient with uncommon mutation had heterogenous response to the TKI. The patients with G719X, S768I, or L861Q respond to TKI. Those patients with exon 20 insertions are mostly insensitive to TKI but sensitive to exon 20 inhibitors.
“Few studies had demonstrated the treatment response to the patients with uncommon EGFR mutation. The patients with G719X, S768I, or L861Q mutations respond to TKI. Those patients with exon 20 insertion are mostly insensitive to TKI [32, 33].”
Kaplan-Meier curve of EGFR wild type, Exon 19 del, L858R, and uncommon mutation disease free survival
Kaplan-Meier curve of EGFR wild type, Exon 19 del, L858R, and uncommon mutation overall survival
References:
- Kwon CS, Lin HM, Crossland V, Churchill EN, Curran E, Forsythe A, Tomaras D, Ou SI: Non-small cell lung cancer with EGFR exon 20 insertion mutation: a systematic literature review and meta-analysis of patient outcomes. Curr Med Res Opin 2022, 38(8):1341-1350.
- Janning M, Suptitz J, Albers-Leischner C, Delpy P, Tufman A, Velthaus-Rusik JL, Reck M, Jung A, Kauffmann-Guerrero D, Bonzheim I et al: Treatment outcome of atypical EGFR mutations in the German National Network Genomic Medicine Lung Cancer (nNGM). Ann Oncol 2022, 33(6):602-615.
Minor points of concern 2: The finding that there are associations of actionable mutations with traits associated with more aggressive disease, but not with clinical outcomes, should be discussed further. What does this mean? Please further relate these finding to other studies and place into a clear and conclusive context.
Response 2: It has been demonstrated that invasive pathological features like lymphovascular invasion and nodal metastasis were associated with poor DFS and OS [32, 33]. In this study we did observe that non-lepidic pattern, pleural invasion, presence of lymphovascular invasion, and higher histologic grading, higher T1 stage (T1a-c), and higher stage IB were significantly more common in EGFR mutation-positive group than those in EGFR mutation-negative group. Nevertheless, only tumor size, lymphovascular invasion, and nodal metastasis were significant factors in multivariate analysis. EGFR mutation or not was not significant factor in the prediction of PFS (p=0.772) and OS (p=0.430). This finding is consistent with recently published French study [17] that EGFR mutations were not associated with recurrence site, disease-free survival, and overall survival in resected stage I to II NSCLC.
References:
- Mordant P Md P, Brosseau S, Milleron B, Santelmo N, Fraboulet-Moreau S, Besse B, Langlais A, Gossot D, Thomas PA, Pujol JL et al: Outcome of Patients With Resected Early-Stage Non-small Cell Lung Cancer and EGFR Mutations: Results From the IFCT Biomarkers France Study. Clin Lung Cancer 2023, 24(1):1-10.
- Ito H, Date H, Shintani Y, Miyaoka E, Nakanishi R, Kadokura M, Endo S, Chida M, Yoshino I, Suzuki H et al: The prognostic impact of lung adenocarcinoma predominance classification relating to pathological factors in lobectomy, the Japanese Joint Committee of Lung Cancer Registry Database in 2010. BMC Cancer 2022, 22(1):875.
- Isaka T, Ito H, Yokose T, Saito H, Adachi H, Murakami K, Miura J, Kikunishi N, Rino Y: Prognostic factors for relapse-free survival in stage IB-IIIA primary lung adenocarcinoma by epidermal growth factor receptor mutation status. BMC Cancer 2022, 22(1):966.
Minor points of concern 3: The T stage in Table 1 is misleading and different from all other descriptions in Tables 1 and 2. The % composition of the individual T stages for a particular subgroup (EGFRm vs EGFRwt) should add up to be 100%, not the combination of the two subgroups for a given stage. With this, the distribution of cases is obscured and bias in the EGFRwt group is present. This should be discussed.
Response 3: Thank you for your comment. We apologize for our mistake and revised the Table 1.
Table 1.
|
Variables |
EGFR(+) n=621 |
EGFR(-) n=413 |
P-value |
|
T stage, n (%) |
|
|
<0.001 |
|
1mi |
23 (4.0) |
68 (16.5) |
|
|
1a |
109 (17.6) |
134 (32.4) |
|
|
1b |
189 (30.4) |
92 (22.3) |
|
|
1c |
123 (19.8) |
34 (8.2) |
|
|
Stage 2 and 3 |
177 (28.5) |
85 (20.6) |
|
Minor points of concern 4: Table 3 has inappropriate comparisons being made in terms of DFS and OS. It is well known increased stage as a result of LN metastases negatively profoundly impacts clinical outcomes in a manner independent from EGFRm status. With this, associations of EGFRm (or EGFRwt) population with clinical outcomes must control for nodal status.
Response 4: Table 3 and table 4 were adjusted. Table 3 now shows the univariate and multivariate analysis of disease-free survival probabilities after surgery. Table 4 now shows the univariate and multivariate analysis of overall survival probabilities after surgery. With univariate analysis, EGFR mutation has impact on the DFS (p=0.023). However, EGFR mutation was not an independent factor (p=0.772) in the multivariate analysis DFS. Multivariate analysis shows that lymphovascular invasion and nodal metastasis were independent factors affecting DFS (p=0.014 and p=0.003) and OS (p=0.026 and p=0.029).
Table 3. Univariate and multivariate analysis of disease-free survival probabilities after surgery
|
|
Univariate analysis |
Multivariate analysis |
||||
|
Variables |
HR |
95% CI |
p-value |
HR |
95% CI |
p-value |
|
Age |
|
|
|
|
|
|
|
≤65 |
|
ref. |
|
|
ref. |
|
|
>65 |
1.788 |
1.240-2.577 |
0.002 |
1.508 |
1.032-2.203 |
0.034 |
|
Sex |
|
|
|
|
|
|
|
Female |
|
ref. |
|
|
ref. |
|
|
Male |
0.833 |
0.571-1.216 |
0.344 |
0.942 |
0.590-1.505 |
0.804 |
|
Smoking |
|
|
|
|
|
|
|
Absent |
|
ref. |
|
|
ref. |
|
|
Present |
1.722 |
1.106-2.683 |
0.016 |
1.251 |
0.710-2.204 |
0.438 |
|
CEA level |
|
|
|
|
|
|
|
≤5 ng/mL |
|
ref. |
|
|
ref. |
|
|
>5 ng/mL |
4.311 |
2.832-6.563 |
< 0.001 |
1.895 |
1.204-2.982 |
0.006 |
|
Tumor size |
|
|
|
|
|
|
|
≤ 20 mm |
|
ref. |
|
|
ref. |
|
|
>20mm |
4.828 |
3.222-7.234 |
<0.001 |
2.494 |
1.574-3.953 |
<0.001 |
|
Histologic predominant subtype |
|
|
|
|
|
|
|
Non-Lepidic |
|
ref. |
|
|
ref. |
|
|
Lepidic |
0.330 |
0.154-0.709 |
0.004 |
0.697 |
0.314-1.549 |
0.376 |
|
Pleural invasion, PI, n (%) |
|
|
|
|
|
|
|
Absent |
|
ref. |
|
|
ref. |
|
|
Present |
3.505 |
2.391-5.136 |
<0.001 |
1.509 |
0.992-2.295 |
0.055 |
|
Lymphovascular invasion |
|
|
|
|
|
|
|
Absent |
|
ref. |
|
|
ref. |
|
|
Present |
4.131 |
2.838-6.014 |
<0.001 |
1.742 |
1.118-2.714 |
0.014 |
|
Lymph node metastasis |
|
|
|
|
|
|
|
N0 |
|
ref. |
|
|
ref. |
|
|
N1 |
5.303 |
3.548-7.924 |
<0.001 |
2.055 |
1.282-3.293 |
0.003 |
|
EGFR mutation |
|
|
|
|
|
|
|
Absent |
|
ref. |
|
|
ref. |
|
|
Present |
1.581 |
1.065-2.346 |
0.023 |
1.964 |
0.698-1.622 |
0.772 |
Table 4. Univariate and multivariate analysis of overall survival probabilities after surgery
|
|
Univariate analysis |
Multivariate analysis |
||||
|
Variables |
HR |
95% CI |
p-value |
HR |
95% CI |
p-value |
|
Age |
|
|
|
|
|
|
|
≤65 |
|
ref. |
|
|
ref. |
|
|
>65 |
1.253 |
0.512-3.067 |
0.621 |
1.596 |
0.618-4.119 |
0.334 |
|
Sex |
|
|
|
|
|
|
|
Female |
|
ref. |
|
|
ref. |
|
|
Male |
1.363 |
0.557-3.336 |
0.497 |
1.392 |
0.500-3.876 |
0.527 |
|
Smoking |
|
|
|
|
|
|
|
Absent |
|
ref. |
|
|
ref. |
|
|
Present |
1.093 |
0.320-3.730 |
0.887 |
1.473 |
0.364-5.955 |
0.587 |
|
CEA level |
|
|
|
|
|
|
|
≤5 ng/mL |
|
ref. |
|
|
ref |
|
|
>5 ng/mL |
1.515 |
0.348-6.594 |
0.580 |
1.904 |
0.408-8.884 |
0.412 |
|
Tumor size |
|
|
|
|
|
|
|
≤ 20 mm |
|
ref. |
|
|
ref |
|
|
>20mm |
3.359 |
1.340-8.420 |
0.010 |
1.385 |
0.441-4.350 |
0.577 |
|
Histologic predominant subtype |
|
|
|
|
|
|
|
Non-lepidic |
|
ref. |
|
|
ref |
|
|
Lepidic |
0.888 |
0.260-3.033 |
0.850 |
0.508 |
0.129-2.000 |
0.332 |
|
Pleural invasion |
|
|
|
|
|
|
|
Absent |
|
ref. |
|
|
ref |
|
|
Present |
4.162 |
1.699-10.197 |
0.002 |
1.895 |
0.654-5.493 |
0.239 |
|
Lymphovascular invasion |
|
|
|
|
|
|
|
Absent |
|
ref. |
|
|
ref |
|
|
Present |
7.329 |
3.034-17.703 |
<0.001 |
3.744 |
1.175-11.937 |
0.026 |
|
Lymph node metastasis |
|
|
|
|
|
|
|
N0 |
|
ref. |
|
|
ref |
|
|
N1 |
8.895 |
3.684-21.473 |
<0.001 |
3.719 |
1.145-12.076 |
0.029 |
|
EGFR mutation |
|
|
|
|
|
|
|
Absent |
|
ref. |
|
|
ref |
|
|
Present |
0.908 |
0.376-2.193 |
0.829 |
0.680 |
0.262-1.770 |
0.430 |
Minor points of concern 4: Figures 2 and 3 should also illustrate the EGFRwt populations.
Response 5: Thank you for your comment. We revised our figures according to your suggestion.
Figure 2 Kaplan-Meier curve of EGFR wild type, Exon 19 del and L858R disease free survival
Figure 3 Kaplan-Meier curve of EGFR wild type, Exon 19 deletion and L858R overall survival

Round 2
Reviewer 2 Report
The revised draft looks acceptable....with exception to their conclusion statement, which overstates their study findings. That needs to be softened.
Author Response
Major points of concern: The revised draft looks acceptable....with exception to their conclusion statement, which overstates their study findings. That needs to be softened.
Response: Thank you for your comment. The conclusion statement was rewrite in the red color.
- Conclusions
In stage I and stage II operable lung adenocarcinoma, our finding suggested that EGFR mutation may be considered as a treatment response predictor for TKI, and may be not a predictor of clinical prognosis. The results of our study should be further validated by other multi-institutional studies.
